# OpenReview forum: "Rapid Response: Mitigating LLM Jailbreaks With A Few Examples"
_ICLR.cc/2025/Conference — Submitted to ICLR 2025_

### Official Review · Reviewer_Y5G6 · 2024-10-30

**Soundness:** 3
**Presentation:** 3
**Contribution:** 2
**Rating:** 5
**Confidence:** 4

**Summary:**

The paper focuses on safeguarding large language models (LLMs) against misuse. Instead of striving for perfect defense, the authors propose rapid response techniques to block entire classes of jailbreaks after observing only a few instances. They introduce a benchmark called RapidResponseBench to test the robustness of defenses. Five rapid response methods using jailbreak proliferation are evaluated, with the strongest method significantly reducing attack success rates by fine-tuning. The study emphasizes the effectiveness of quick adaptation to new jailbreaks to mitigate LLM misuse.

**Strengths:**

1. This is an important and interesting topic.
2. Some findings are particularly noteworthy. For instance, Guard Fine-tuning and Regex significantly reduce the attack success rate with just a single example from each jailbreak strategy.
3. Overall, the experiments are sound.

**Weaknesses:**

1. The main concern is that the threat model is overly favorable to the defender, who has access to a few examples from each jailbreak strategy. This allows the defender to develop any adaptive method to counter the attacks.
2. The novelty and technical contribution of this paper are limited, as it primarily evaluates existing attack and defense methods. Nonetheless, I appreciate the effort put into this work.
3. Lacking the discussion of adaptive attacks.
4. In Section 3.3, the authors examine the role of jailbreak proliferation in rapid response. However, as Figure 3 illustrates, apart from Guard Fine-tuning, jailbreak proliferation does not significantly impact the effectiveness of the rapid response defense. Despite this, the authors claim that jailbreak proliferation is crucial to its effectiveness—a conclusion not supported by the evidence.

**Questions:**

Please see comments.

---

### Official Review · Reviewer_85ci · 2024-11-02

**Soundness:** 3
**Presentation:** 3
**Contribution:** 3
**Rating:** 5
**Confidence:** 4

**Summary:**

In this paper, the authors introduce a new paradigm in safety of LLM, focusing on a reactive approach to mitigate jailbreaking attacks on LLMs. The authors propose a benchmark called RapidResponseBench, evaluating the efficacy of rapid response methods that adapt to a limited number of observed jailbreak attempts. They examine five rapid response techniques that involve jailbreak proliferation, generating additional jailbreak examples based on a few observed attacks. The study finds that fine-tuning an input classifier on proliferated jailbreaks notably reduces the attack success rate, demonstrating the promise of a reactive defense model for LLM safety.

**Strengths:**

1. The authors introduce a new approach to LLM jailbreak mitigation. Instead of creating robust defenses upfront, it offers a flexible response mechanism that adapts to emerging jailbreak strategies.

2. This paper is well-structured, detailing five different rapid response methods and meticulously evaluating each on in-distribution and out-of-distribution attack success rates.

**Weaknesses:**

1. While rapid response methods perform well with observed examples, their efficacy on truly novel attack types remains uncertain, particularly for out-of-distribution attacks that deviate substantially from observed jailbreak patterns.

2. Some rapid response techniques lead to increased refusal rates on benign queries, impacting user experience negatively. This side effect raises questions about balancing defense effectiveness with accessibility for non-malicious users.

3. The success of the defense relies heavily on the quality and diversity of proliferated examples. Limited generalizability of proliferated examples may reduce the defense's effectiveness if attackers adopt radically different jailbreak techniques.

**Questions:**

1.	 How does the approach perform with jailbreak strategies that differ fundamentally from the training set, especially in contexts where adversaries evolve new types of jailbreaks?

2.	Could additional tuning techniques further mitigate increased refusal rates for benign queries? What are some potential ways to balance refusal rates with rapid response efficacy?

3.	Given the reliance on jailbreak proliferation, how does the quality of proliferated examples influence effectiveness? Have different language models been tested for proliferation to ensure robustness across model variations?

---

### Official Review · Reviewer_sen4 · 2024-11-04

**Soundness:** 3
**Presentation:** 3
**Contribution:** 2
**Rating:** 8
**Confidence:** 3

**Summary:**

The paper propose a rapid response technique that aims to block whole classes of jailbreaks after observing only a few instances, rather than seeking perfect adversarial robustness.

The authors present a new benchmark called `RapidResponseBench` t assess the effectiveness of this strategy. The benchmark evaluates five rapid response methods (Regex, Guard Fine-tuning, Embedding, Guard Few-shot and Defense Prompt), all utilizing jailbreak proliferation, which generates additional jailbreaks similar to observed examples. The authors also consider in-distribution and out-of-distribution jailbreaks, as well as refusal of benign queries.

The results highlight the potential of rapid response to adaptively strengthen defenses in real time against emerging jailbreak methods.

**Strengths:**

1. The paper is clearly written, with comprehensive descriptions of each attack type, defense strategy, and evaluation metric.

2. The new paradigm introduced in this paper, `Jailbreak Rapid Response`, is a significant departure from traditional adversarial robustness approaches. Given the rapid development of LLM research, the concept of responding rapidly to jailbreaks using limited examples is innovative and addresses a critical need in LLM security.

3. The authors promise in Section 7 that the benchmark will be open-sourced, making it easy for others to follow and implement.

**Weaknesses:**

1. The effectiveness of the rapid response methods heavily relies on the quality and quantity of proliferated examples. As seen in Figure 3, different proliferation models and varying proliferation attempts influence the final outcomes. Therefore, further exploration into the quality and quantity of proliferated examples, such as using different proliferation templates, could enhance the results.

2. The paper examines the effectiveness of five existing rapid response methods against six types of attacks. However, Figure 5 shows that different defense methods have certain limitations against specific attacks; for instance, Guard Few-shot shows no improvement against Cipher even as the number of observed jailbreaks increases. While this may suffice for a benchmark, I suggest the authors consider proposing their new rapid response method based on their findings, or combining different methods to improve defense against jailbreaks.

3. Since the proposed `rapid response technique` emphasizes real-world deployment, the authors might also  discuss practical challenges in deploying rapid response systems, such as computational costs and real-time detection efficiency of specific defense strategies.

**Questions:**

1. In Figure 3 (right), the study investigates the impact of varying the number of proliferation attempts per jailbreak strategy, but the x-axis is incorrect since the proliferation attempts are not evenly spaced. I recommend adjusting the figure to accurately reflect the true trend.

---

### Official Review · Reviewer_fFpP · 2024-11-04

**Soundness:** 2
**Presentation:** 3
**Contribution:** 2
**Rating:** 5
**Confidence:** 4

**Summary:**

This paper tries to defend against jailbreak attacks by observing jailbreak instances from the attack and updating the defense using a response strategy. By observing as few as 1 attack instances, the response strategy is able to reduce attacks success rate by employing LLM guided proliferation to synthetically generate more attacks instances. The paper considers 6 attacks and their variants, as well as 5 types of defenses.

**Strengths:**

1. The paper approaches a very important and unexplored direction of post hoc response strategy to jailbreak attacks.
2. I appreciate the author's attempt at evaluating against out of distribution variants of the attacks.
3. The paper is well written and easy to follow.

**Weaknesses:**

1. **Incorrect way of reporting results**: Plotting the results as an average of three very different models can be misleading. The authors should show results for these models separately.
2. **Considered attacks are not adaptive enough**: When evaluating any defense, one needs to account for an adaptive adversary (preferably one that knows the details of the defense in place). Although, the authors attempt to account for minor variations in the attacks, it is not enough. Instead of a fixed minor variant, an adaptive adversary can always keep trying variants until one succeeds. The highlights the importance of an automated adaptive adversary. For instance, an adaptive adversary against Guard Fine-tuning could simply add an universal perturbation to the input prompt to bypass the guard model [1].
3. **Missing other automated black box attacks**: While evaluating a defense, it is necessary to consider automated attacks as they are adaptive by design (although they can be further modified to account for the defense).  The authors evaluate the automated attack PAIR, but other well developed black box automated attacks are missing. [2,3,4,5]
4. **Missing eval against ensemble of attacks**: For the 6 attacks considered in the paper, the defenses / response strategies need to be evaluated against an ensemble of all the attacks at once. Since, an adversary can simply choose whatever attack works, and the defense needs to be simultaneously defend against all attacks.

[1] Mangaokar, Neal, et al. "Prp: Propagating universal perturbations to attack large language model guard-rails." arXiv preprint arXiv:2402.15911 (2024).
[2] Hayase, Jonathan, et al. "Query-based adversarial prompt generation." arXiv preprint arXiv:2402.12329 (2024).
[3] Mehrotra, Anay, et al. "Tree of attacks: Jailbreaking black-box llms automatically." arXiv preprint arXiv:2312.02119 (2023).
[4] Sitawarin, Chawin, et al. "Pal: Proxy-guided black-box attack on large language models." arXiv preprint arXiv:2402.09674 (2024).
[5] Andriushchenko, Maksym, Francesco Croce, and Nicolas Flammarion. "Jailbreaking leading safety-aligned llms with simple adaptive attacks." arXiv preprint arXiv:2404.02151 (2024).

**Questions:**

1. It is unclear whether the PAIR attack's optimization is run again (or the previous attack instances are used as it is) after the response strategy is applied.
2. Were there any refusals from the proliferation model while generating synthetic jailbreaks?

---

### Meta-Review · Area_Chair_DFkC · 2024-12-21

**Metareview:**

This paper investigates rapid response techniques aimed at blocking entire categories of jailbreaks after observing only a small number of attacks. It introduces RapidResponseBench to evaluate a defender's ability in this setting. Overall, the paper is borderline, with mixed reviews.

The primary reason the reviewers support this work is its novel approach of using rapid response mechanisms to defend against jailbreak attacks—this is a new setting. The benchmark proposed will be valuable to the research community.

After a discussion among the AC and the reviewers, two main concerns emerged: (1) the limited generalizability and (2) the symmetry introduced by the threat model, which inherently favors the defender. Moreover, the reviewers note that the current experimental results do not fully support the authors’ claim that rapid response mechanisms can effectively mitigate novel jailbreak methods.

AC agrees with the reviewers’ concerns and hopes the authors can further refine this paper.

**Additional Comments On Reviewer Discussion:**

Two reviewers are actively involved in the rebuttal process, and all reviewers agree that the setting is novel and useful for the community. However, during the rebuttal, reviewer Y5G6 remained concerned about generalization and thought the current defense inherently favors the defender. After the rebuttal, other reviewers expressed similar concerns about this point.

---

### Decision · Program_Chairs · 2025-01-22

Reject